# The *Drosophila mauritiana* synaptonemal complex protein C(3)G repatterns the recombination landscape of *Drosophila melanogaster*

Stacie E. Hughes[1]*, Cynthia Staber[1], Grace McKown[1], Zulin Yu[1], Justin P. Blumenstiel[2], R. Scott Hawley[1]†

1 Stowers Institute for Medical Research, Kansas, Missouri, United States of America, 2 Department of Ecology and Evolutionary Biology, University of Kansas, Lawrence, Kansas, United States of America

† Deceased
* sfh@stowers.org

## Abstract

Meiotic recombination plays an important role in ensuring proper chromosome segregation during meiosis I through the creation of chiasmata that connect homologous chromosomes. Recombination plays an additional role in evolution by creating new allelic combinations. Organisms display species-specific crossover patterns, but how these patterns are established is poorly understood. *Drosophila mauritiana* displays a different meiotic recombination pattern compared to *Drosophila melanogaster*, with *D. mauritiana* experiencing a reduced centromere effect, the suppression of recombination emanating from the centromeres. To evaluate the contribution of the synaptonemal complex (SC) C(3)G protein to these recombination rate differences, the *D. melanogaster* allele was replaced with *D. mauritiana* *c(3)G* coding sequence. We found that the *D. mauritiana* C(3)G could interact with the *D. melanogaster* SC machinery to build full length tripartite SC and chromosomes segregated accurately, indicating sufficient crossovers were generated. However, the placement of crossovers was altered, displaying an increase in frequency in the centromere-proximal euchromatin indicating a decrease in the centromere effect, similar to that observed in *D. mauritiana* females. Recovery of chromatids with more than one crossover was also increased, likely due to the larger chromosome span now available for crossovers. As replacement of a single gene mediated a strong shift of one species' crossover pattern towards another species, it indicates a small number of discrete factors may have major influence on species-specific crossover patterning. Additionally, it demonstrates the SC, a structure known to be required for crossover formation in many species, is likely one of these discrete factors.

**Data availability statement:** Original data underlying this manuscript can be accessed from the Stowers Original Data Repository at http://www.stowers.org/research/publications/libpb-2555.

**Funding:** Funding for this work came from the Stowers Institute for Medical Research (R.S.H) and a National Science Foundation award 2025197 (J.P.B.). The funders had no role in study design, data collection and analysis, decision to publish, or preparation of the manuscript.

**Competing interests:** The authors have declared that no competing interests exist.

## Author summary

Meiotic crossovers are important for ensuring proper chromosome segregation and generating genetic diversity. Different species display unique crossover patterns but the mechanisms that establish these patterns are poorly understood. The synaptonemal complex (SC) is built between meiotic chromosomes and promotes crossover formation. Replacement of the SC gene *c(3)G* in the fruit fly *Drosophila melanogaster* with *Drosophila mauritiana c(3)G* resulted in full-length SC assembly and proper chromosome segregation. However, the *D. melanogaster* crossover pattern was shifted to appear more similar to *D. mauritiana*. This demonstrates that crossover patterning can be influenced by minor changes in the makeup of the SC.

## Introduction

Meiosis is a specialized cell division that results in haploid gametes. In many organisms, there are several important steps that must occur for meiosis to be successful. Homologous chromosomes pair, double-strand breaks (DSBs) are induced, chromosomes undergo synapsis, and a subset of the DSBs are repaired into crossovers that physically link the homologous chromosomes throughout prophase. These physical linkages, known as chiasmata, act in conjunction with cohesion between the sister chromatids and spindle microtubules to ensure homologs properly orient and segregate towards opposite spindle poles during the meiosis I division. Failure to generate a crossover on each chromosome can lead to the missegregation of the homologous chromosomes and the production of aneuploid gametes that cause miscarriage, birth defects, and infertility [1,2].

While a single crossover per chromosome can ensure proper segregation at the meiosis I division, many organisms make more than one crossover per chromosome. For example, *Drosophila melanogaster* females make an average of one crossover per chromosome arm, which results in two crossovers per autosome, and yeasts, such as *Saccharomyces cerevisiae,* make multiple crossovers per chromosome [3–5]. The honey bee, *Apis mellifera*, has a very high level of meiotic recombination, with more than five recombination events per chromosome pair per meiosis [6]. Additionally, organisms differ in how crossovers are distributed across the chromosomes with some chromosomal regions more or less likely to experience a crossover. For example, in many mammals, recombination preferentially occurs near "recombination hotspots" that are dependent on PRDM9 (reviewed in [7]). Conversely, recombination is suppressed in the repetitive heterochromatin in most species [8–12]. These differences in crossover number and placement lead to species-specific recombination patterns that can differ even between relatively closely related species [3,13–15]. These differences can potentially influence evolution because recombination rate and placement affect the likelihood that alleles will be inherited together in offspring. Regions of low recombination may accumulate deleterious genetic elements, such

as transposable elements, due to Muller's Ratchet, the accumulation of deleterious variants due to the stochastic loss of the most fit haplotype [16]. Recombination, by uncoupling linked alleles, provides an opportunity to eliminate, from a population, less fit alleles that are linked to beneficial alleles residing on the same chromosome (reviewed in [17,18]). Since crossover rate and placement can affect the capacity of a species to adapt to changing conditions, it is important to understand the factors that can influence crossover patterning.

The fruit fly *Drosophila melanogaster* has long been a model for studying aspects of crossover patterning, such as crossover interference (the designation of one crossover inhibiting the designation of another crossover nearby) and the centromere effect [4,19,20]. The centromere effect is the polar suppression of recombination emanating from the centromeres (reviewed in [10,21,22]). Centromeres are the location of kinetochore assembly that are identified by the histone H3 variant centromere protein A (CID in *Drosophila*) and in *D.melanogaster* map to regions of heterchromatic repeats, which are also regions where recombination is suppressed [23]. The centromere-proximal heterochromatin is thought to act as a buffer, functioning as an inert spacer, against the centromere effect in *D. melanogater* rather than directly providing the mechanism for the crossover suppression in the centromere-proximal euchromatin. In experiments that move centromere proximal heterochromatin away from the centromere, thus bringing euchromatic sequences closer to the centromere, those centromere-proximal euchromatic regions experience decreased levels of crossing over [24–26]. These experiments indicate the centromere effect is mediated by the centromere itself or trans-acting factors rather than the surrounding heterochromatin. While these experiments ruled out the role of the heterochromatin in mediating the centromere effect, they were not designed to distiniguish whether the centromere effect was mediated by *trans*-acting factors, *cis*-acting elements within the centromere, or a combination of the two.

Interestingly, both True et al. [13] and Hawley et al. [27] found that the centromere effect was lessened in the closely related species *Drosophila mauritiana*. *D. melanogaster* and *D. mauritiana* display a difference in crossover number and patterning despite their relatively close evolutionary history, similar genome sizes, and similar amounts of pericentric heterochromatin [13,27]. *D. mauritiana* has an increased total map distance of 1.8-fold (463 vs. 259 cM) according to True *et al.* [13] and 1.37-fold (379 vs 280 cM) for Hawley et al.[26,27] compared to *D. melanogaster*. In both studies, the difference between *D. mauritiana* and *D. melanogaster* was not uniform between chromosome arms. The difference in total map distance is mostly explained by an increase in crossing over in euchromatic regions proximal to the centromeres in *D. mauritiana* compared to *D. melanogaster* demonstrating that *D. mauritiana* has a decreased centromere effect compared to *D. melanogaster* [13,27]. The decreased centromere effect would increase the span available to acquire a crossover on most chromosome arms allowing for the increased double crossover chromatids and total map length in *D. mauritiana* [13,27].

The mechanisms controlling crossover patterning have only just begun to be elucidated and even less understood is how species-specific patterns evolve. Are changes in crossover patterning the result of small changes in many genetic loci, or can global patterning be shifted by changes in just a few genes? One *trans*-acting factor that appears to mediate the differences in crossover patterning between *D. melanogaster* and *D. mauritiana* is Mei-218 [28]. Mei-218 is a pro-crossover protein and part of the Mei-MCM complex [29]. When the *D. mauritiana* version of *mei-218* was expressed as a transgene in *D. melanogaster* females that were mutated at the endogenous *mei-218* locus, the crossover pattern of the *D. mauritiana mei-218* expressing flies displayed an increase in total crossing over compared to the flies expressing a *D. melanogaster mei-218* transgene [28]. However, both transgenes displayed a pattern that was somewhat distinct from wild-type *D. melanogaster* flies indicating *mei-218* is unlikely to be the sole cause of differences in meiotic recombination between these two species. As crossover patterns are complex there may be other factors that majorly influence crossover patterning.

A meiotic structure that may also influence meiotic crossover patterning is the synaptonemal complex (SC). By electron microscopy the SC is a ladder-like structure that builds along the homologous chromosome during early pachytene. It has a tripartite structure with two lateral elements (LEs) that interact with the homologous chromosomes. The distance

between the LEs is spanned by transverse filament proteins that are stabilized by additional proteins within the central region of the SC. Fig 1A shows the current model of the *D. melanogaster* SC. C(3)G is the *Drosophila* major transverse filament protein that spans across the SC and is the functional ortholog to Zip1 in yeast and SCP1 in mammals (reviewed [30,31]). C(3)G is modeled to be a homodimer with the C-termini of the homodimers interacting with the LEs [32–34]. The N-termini of the C(3)G dimers interacts with homodimers from the opposing LE at the central element of the SC. The central region proteins Corona (Cona) and Corolla are thought to stabilize C(3)G assembly since absence of any of the three proteins result in a failure to assemble SC [32,35,36].

In many species, the SC is required for crossover formation (reviewed [31,37]). While the physical structure of SC appears conserved between divergent species based on electron microscopy, the individual proteins of the SC are rapidly evolving at the amino acid level. SC proteins can be difficult to identify using homology to amino acid sequence alone [38–41]. Because the SC proteins are rapidly evolving and involved in crossover formation, they have the potential for being another avenue for regulating crossover patterning between species. It has been shown previously that conditions that perturb the SC can result in changes in crossover number and location [12,42–44]. For example, in *C. elegans,* partial depletion of a SC component led to increases in COSA-1 foci, which mark crossovers, and a decrease in crossover interference [42]. In *D. melanogaster,* in-frame deletions within the coiled-coil domain of the transverse filament protein C(3)G resulted in SC that disassembled earlier than wild type, coinciding with a shift of crossovers towards the centromere-proximal euchromatin [43]. In *Arabidopsis thaliana* the SC appears non-essential for the formation of crossovers, but loss of the SC leads to increased crossover formation, suggesting the role of the SC is regulating crossover interference in this species [45]. Additionally, current models propose that pro-crossover factors diffuse within the SC, and DSBs are selected to become crossovers through accumulation of pro-crossover factors by a coarsening process [46–49], indicating that changes in SC composition have the potential to alter coarsening dynamics, and ultimately, the number and position of crossovers.

To more directly test the role of SC components in crossover patterning and the impact that SC sequence divergence has on recombination, we tested whether the replacement of the *D. melanogaster* SC protein C(3)G with the *D. mauritiana* version of C(3)G could change crossover patterning. The *D. melanogaster* C(3)G protein shares 89.1% amino acid identity with the *D. mauritiana* C(3)G protein. We show that replacing the *D. melanogaster* c(3)G with the *D. mauritiana* version led to changes in the crossover patterning of *D. melanogaster*, causing it to more closely resemble the *D. mauritiana* pattern, with an increase of crossovers in the centromere-proximal euchromatin and an increase in the recovery of double crossover chromatids. These results suggest that the difference in crossover patterning between *D. melanogaster* and *D. mauritiana* and the centromere effect in *D. melanogaster* may be largely regulated by a small number of *trans*-acting factors, including C(3)G. In addition, our findings demonstrate that the rapid evolution of the SC proteins is likely to have functional consequences for establishing crossover patterning.

## Results

We sought to determine whether the rapidly evolving SC protein C(3)G might contribute to differences in the recombination landscape between *D. mauritiana* and *D. melanogaster*. As the transverse filament protein C(3)G is the major determinant of SC width [33,43] and distance between the homologs, this protein has a great potential for influencing crossover patterning. The C(3)G proteins in *D. mauritiana* and *D. melanogaster* share 89.1% identity, with the amino acid changes dispersed along the length of the protein (S1 and S2 Figs). The level of amino acid difference between the species suggests the *D. mauritiana* C(3)G could potentially be structurally similar enough to interact with the other *D. melanogaster* proteins of the SC to allow for proper SC assembly but has enough amino acid difference that subtle differences in SC function might be found. To directly test whether C(3)G plays a role in the crossover patterning difference between the two species, the gene body of the *D. melanogaster* c(3)G gene was replaced with a codon-optimized coding region of the *D. mauritiana c(3)G* gene by WellGenetics using CRISPR-Cas9 technology (see Materials and methods). This replacement

A

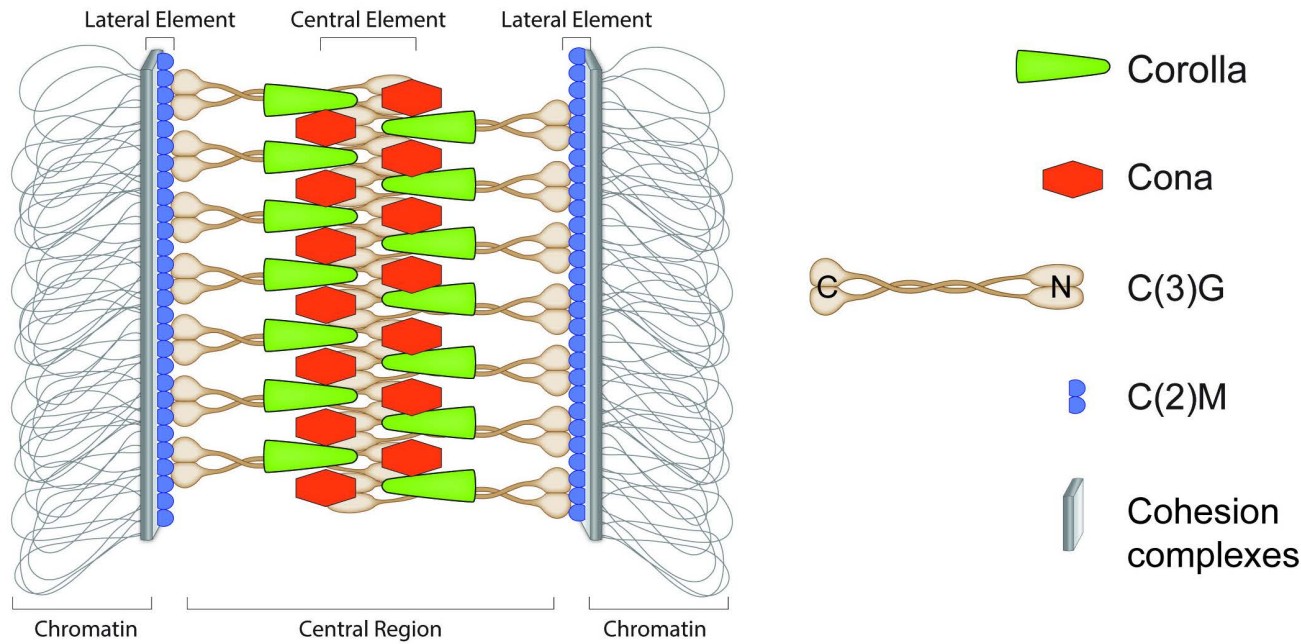

B

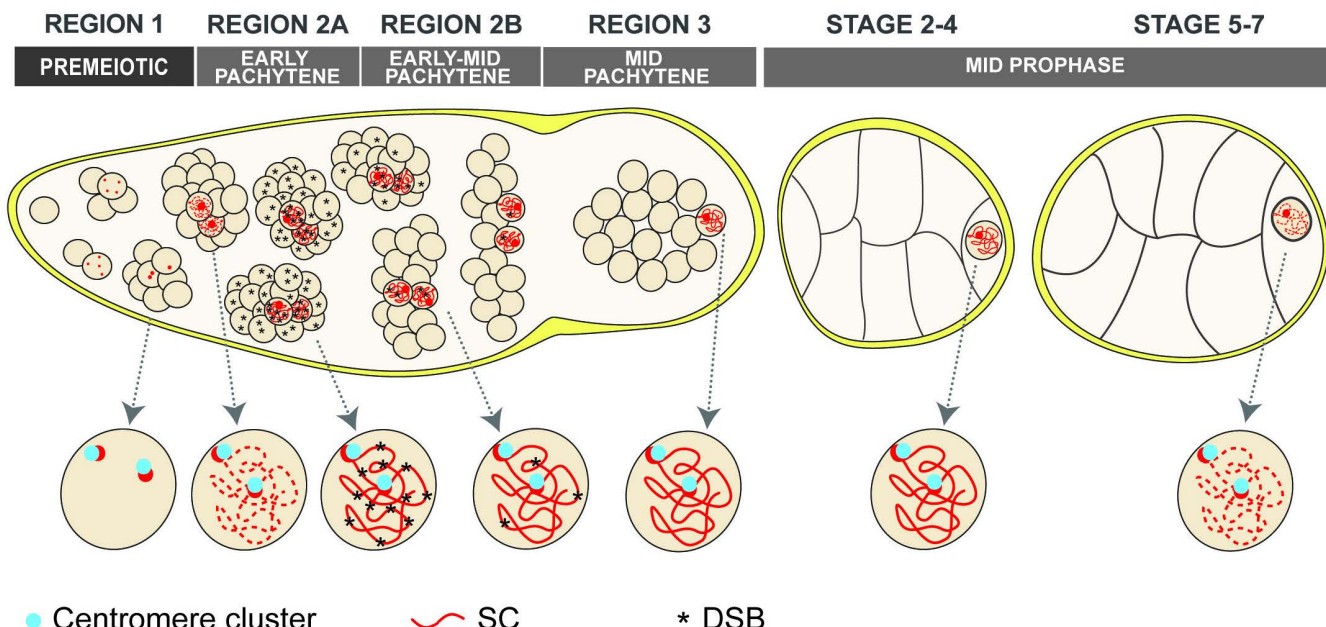

**Fig 1. Models of SC structure and dynamics in _D. melanogaster_ females.** (A) Model of the _D. melanogaster_ SC. Cohesin complexes (represented as a gray bar) interact with the homologous chromosomes. The protein C(2)M (blue partial circles) is present along the chromosomes at the LE. Dimers of the transverse filament protein C(3)G (brown) in the central region connect to the LEs at the C-terminal end of the dimers and the N-terminal part of the dimers interact with opposing C(3)G dimers. C(3)G is stabilized by the proteins Cona (orange hexagons) and Corolla (green triangles) in the central region. (B) Diagram of the meiotic events during the early developmental stages of the ovary. Individual nuclei are shown enlarged below the illustration

of development. In the premeiotic region 1, the cystoblast divides four times to make a 16-cell interconnected cyst. SC proteins (red) begin to load near the centromeres (blue circles in the enlarged nuclei) that begin to cluster during the mitotic divisions. At the entry to region 2A (early pachytene), SC components load at additional sites along the chromosome arms. In region 2A the SC is full-length in 2-4 nuclei of each cyst, DSBs (asterisks) are induced in the cysts, and the centromeres are clustered into an average of two clusters. As the cysts progress to region 2B (early-mid pachytene), DSBs become progressively repaired based on γH2Av and two SC-positive nuclei are present within the cysts. By region 3 (mid-pachytene) the pro-oocyte is the only nucleus with full-length SC and the γH2AV marker for DSBs is absent. Centromeres remain clustered to an average of two foci. After the egg chamber buds off from the germarium the SC progressively disassembles as the egg chamber continues development.

is termed *c(3)G^mau*. In the *c(3)G^mau* flies, all other genes encoding central region and LE components of the SC still encode the endogenous *D. melanogaster* versions.

With the sequence divergence of C(3)G between the two species, there was the possibility that the C(3)G^mau protein would not be able to fully function within the context of building an SC structure with the *D. melanogaster* versions of the other SC components. We first examined SC assembly by immunofluorescence in *c(3)G^mau* ovaries compared to a control line with the same genetically marked chromosomes but with the endogenous *D. melanogaster c(3)G* (*c(3)G⁺*). In *D. melanogaster* wild-type ovaries, central region components of the SC first load near the centromeres during the mitotic divisions that form the 16-cell interconnected germline cyst (Fig 1B) (reviewed in [50]). As the cyst enters meiosis, SC components load at additional sites along the chromosome arms (zygotene). By region 2A (early pachytene), multiple nuclei of the cyst have formed full-length SC along the length of the chromosomes. As the cyst progresses to region 2B (early-mid pachytene), the SC progressively begins to disassemble from all but the nucleus selected to become the pro-oocyte, resulting in one nucleus with full-length SC by region 3 (mid-pachytene). After the cysts buds off to form the egg chamber, the SC in the pro-oocyte progressively disassembles [51]. Based on an antibody recognizing the central region protein Corolla, SC assembly in regions 2A and 2B (early to early-mid pachytene) in *c(3)G^mau* ovaries looked highly similar to *c(3)G⁺* with long continuous tracts of SC in numerous nuclei, indicating the C(3)G^mau protein could support full-length SC assembly with the other *D. melanogaster* SC proteins (Fig 2A and 2B). The monoclonal antibody raised against the C-terminus of *D. melanogaster* C(3)G fails to recognize the *D. mauritiana* C(3)G (S3 Fig), so a polyclonal antibody raised against 324 amino acids in the C-terminal half of the *D. mauritiana* C(3)G protein (mau C(3)G) was generated (see Materials and methods). The mau C(3)G antibody displayed a localization pattern similar to Corolla in the *c(3)G^mau* replacement line, and it localized like Corolla in *c(3)G⁺* germaria expressing *D. melanogaster* C(3)G (Fig 2A and 2B). This indicates that the mau C(3)G antibody recognizes the C(3)G protein from both species.

To verify if the SC in *c(3)G^mau* germaria has a tripartite structure, the localization of Corolla and mau C(3)G antibodies were examined by stimulated emission depletion (STED) microscopy. In both *c(3)G⁺* and *c(3)G^mau* germaria, the fluorescence for the mau C(3)G antibody could be observed to split into two parallel tracks as has been observed previously by super-resolution microscopy for the *D. melanogaster* C-terminal antibody in wild-type ovaries (Fig 3A and 3B) [36]. In both *c(3)G⁺* and *c(3)G^mau* germaria, Corolla was located between the two tracks of C(3)G fluorescence, indicating that *c(3)G^mau* assembles the expected tripartite SC structure. The distance between the two tracks of C(3)G fluorescence was measured to determine if the dimensions of the SC were altered (Fig 3C). The average distance between the tracks of mau C(3)G antibody fluorescence in *c(3)G^mau* was 112.1 nm (SD +/- 8.7 nm, N (measurements) =25), while the distance between the mau C(3)G antibody staining in *c(3)G⁺* ovaries was significantly larger at 127.0 nm (SD +/- 11.6, N = 27; p < 0.0001). The super-resolution microscopy indicates the C(3)G^mau protein can function with the *D. melanogaster* LE and central element proteins to promote full-length tripartite SC assembly. The amino acid differences in the C(3)G^mau protein do not suggest an obvious explanation in the change in width compared to SC in *c(3)G⁺* ovaries. As 112.1 nm is within the width 90–150 nm range cited for SC in other organisms, it is unclear what functional consequence, if any, this difference in SC width would have on the functions of the SC.

While tripartite SC is assembled in early pachytene in *c(3)G^mau* ovaries, the stability of the full-length SC could be affected. We next examined SC dynamics within the germarium and observed full-length SC for the entirety of region 2A

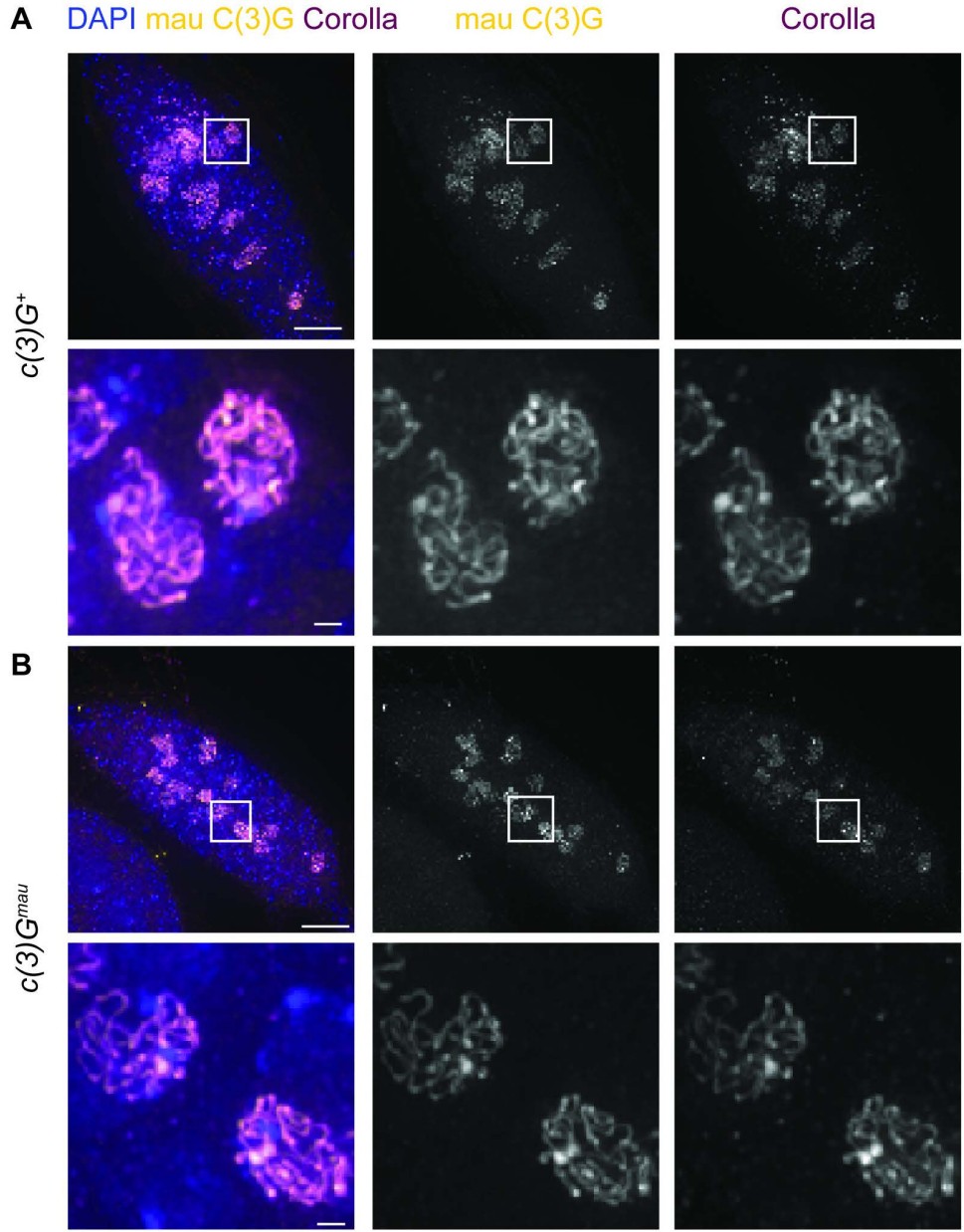

**Fig 2. Full length SC assembles in *c(3)Gmau* ovaries.** (A-B) Shown are antibodies raised against *D. melanogaster* Corolla (magenta) and the C-terminal end of *D. mauritiana* C(3)G (mau C(3)G) (yellow). DAPI is shown in blue. Boxed regions are shown enlarged. (A) In *c(3)G⁺* the mau C(3)G antibody appears as full-length tracks that overlap with the Corolla immunofluorescence indicating the mau C(3)G antibody recognizes *D. melanogaster* C(3)G. (B) In *c(3)Gmau* both mau C(3)G and Corolla antibodies appear as full-length tracks in early to early-mid pachytene similar to *c(3)G⁺* indicating SC assembly occurs normally in *c(3)Gmau*. Scale bar = 10 μm in germaria and 1 μm in enlarged images. Images are projections from z-stacks.

(early pachytene) where DSBs are normally initiated in *c(3)Gmau* ovaries (Fig 4A and 4B). The SC continued to appear full-length in the majority of germaria in region 2B (early-mid pachytene), which is the region DSBs continue to undergo repair and crossovers become designated based on previous data [52,53]. The *c(3)G⁺* retained full-length SC in the nucleus selected to become the oocyte at region 3 (mid pachytene), consistent with previous studies (Fig 4) [43,51]. In region 3 of

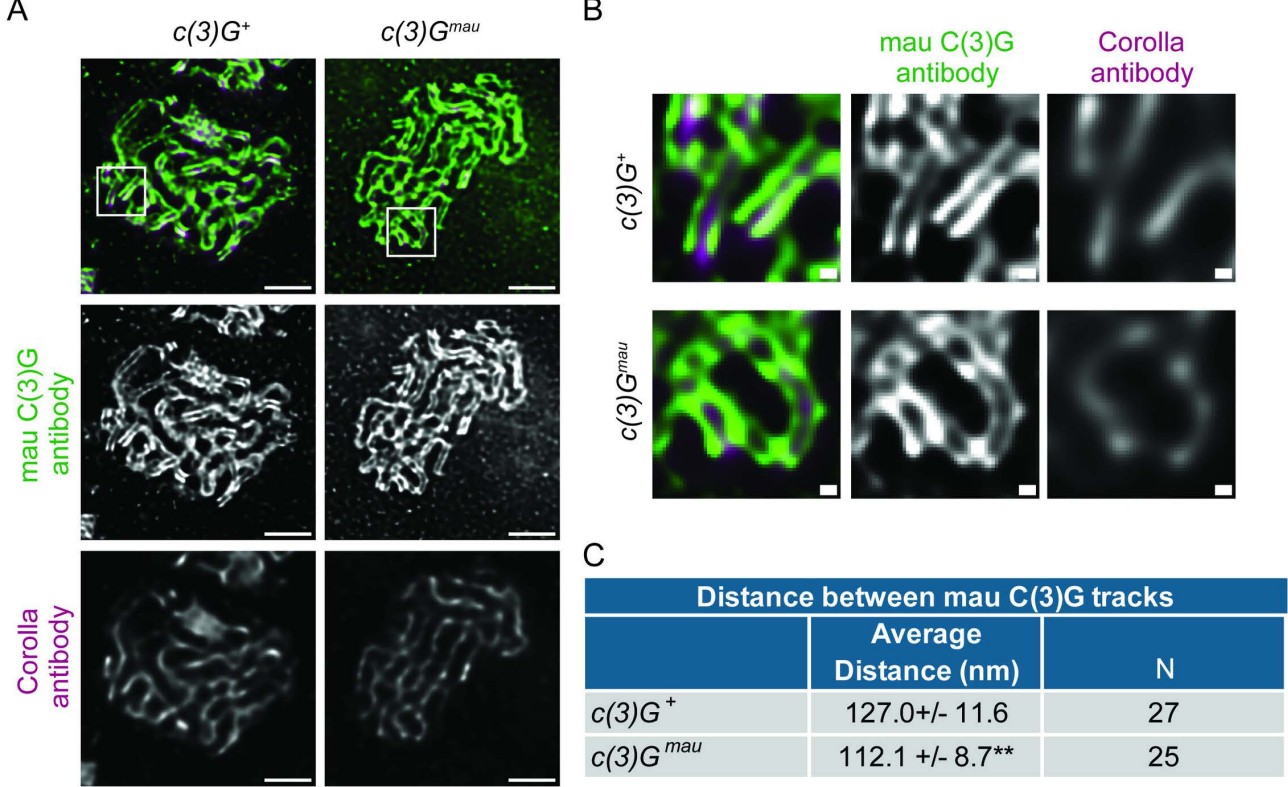

**Fig 3. The SC in *c(3)G^mau* ovaries has a tripartite structure but is narrower than SC from *c(3)G^+* ovaries.** (A) Nuclei from *c(3)G^+* and *c(3)G^mau* acquired using STED microscopy. The mau C(3)G antibody against the C-terminal part of the protein can be resolved as two tracks (green) in both genotypes with an antibody to the central region protein Corolla localized between the tracks (magenta). Scale bar = 1 μm. Boxed region is magnified in (B). Scale bar = 0.1 μm. (C) Average and standard deviation are presented for measurements taken between the two tracks using the mau c(3)G antibody. N = number of measurements. Statistically significant from *c(3)G^+* at **p < 0.01 by Mann-Whitney U test.

*c(3)G^mau* germaria, a mild early SC disassembly phenotype was observed in some oocytes, with approximately a quarter of the oocyte nuclei displaying some discontinuities in the SC tracks (Fig 4A and 4B). The fragmentation phenotype was mild, in that most of these nuclei displayed a combination of long tracks of SC and some SC fragments. This mild onset of SC disassembly indicates the *C(3)G^mau* containing SC may be less stable or its disassembly is being regulated differently compared to SC in *c(3)G^+* germaria (Fig 4). Based on cytology, C(3)G^mau can function well enough with the *D. melanogaster* LE and central element components to support the formation of full-length tripartite SC in early pachytene despite the amino acid differences. The mild early SC disassembly phenotype suggests the amino acid changes in C(3)G^mau may lead to a protein that functions less well in maintaining SC through all stages of meiosis, compared to its *D. melanogaster* counterpart.

## C(3)G^mau protein supports wild-type levels of centromere clustering and chromosome segregation

We next examined the ability of C(3)G^mau to function in processes that depend on SC proteins. Immunofluorescence studies in *D. melanogaster* using an antibody recognizing the *Drosophila* homolog of CENPA, Centromere Identifier (CID) have demonstrated that SC components are required for the clustering of homologous centromeres during early pachytene, with an average of two foci per oocyte, [54,55]. Null mutations of SC central region components in *D. melanogaster* result in a failure in centromere clustering [36,55]. The number of CID foci was scored in meiotic nuclei at different stages

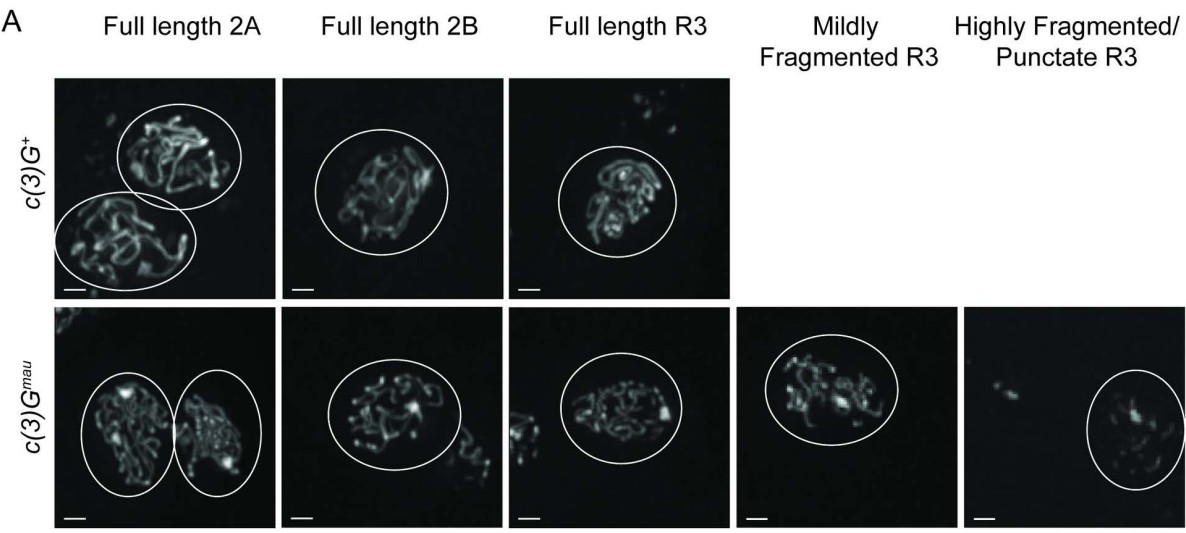

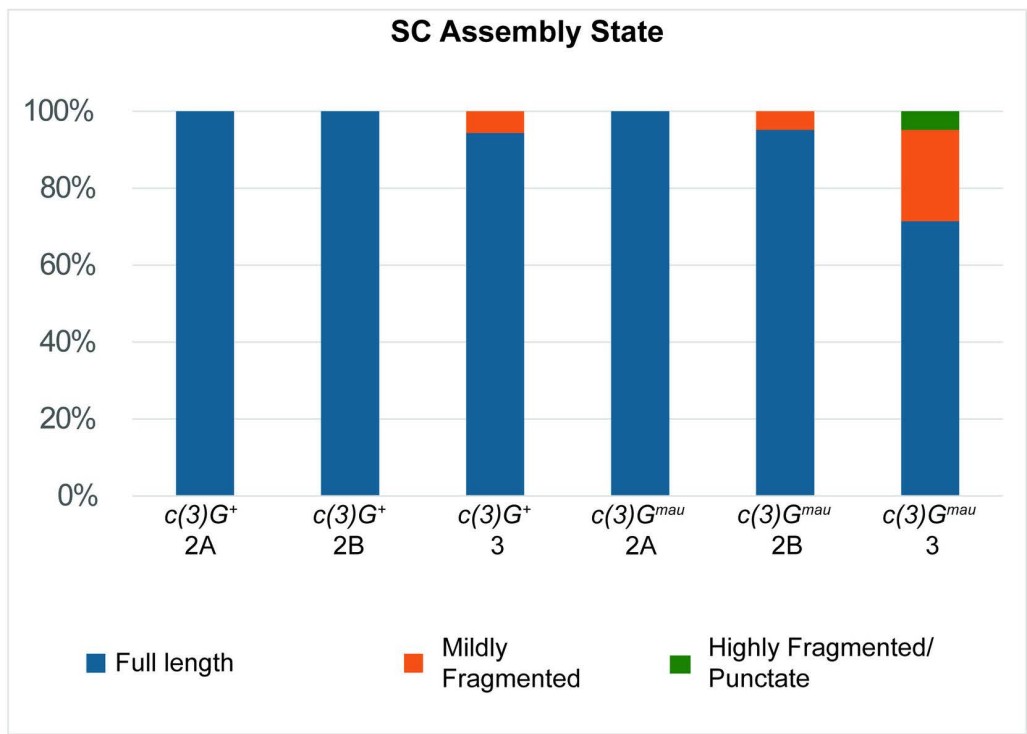

**Fig 4. SC fully assembles in region 2A (early pachytene) but begins to disassemble early in some *c(3)Gmau* ovaries.** (A) Examples of nuclei from *c(3)G⁺* and *c(3)Gmau* ovaries at regions 2A, 2B, and 3. Shown are the full-length SC observed with the mau c(3)G antibody in the majority of nuclei at these stages along with an example of the mildly fragmented, or highly fragmented SC scored at region 3 in *c(3)Gmau* ovaries. SC was labeled with the mau C(3)G antibody. Images are projections from partial z-stacks and scale bar = 1 μm. (B) Shown is the percentage of germaria with the scored SC assembly state for *c(3)G⁺* (N = 18 germaria) and *c(3)Gmau* (N = 21 germaria) for germarium regions 2A (early pachytene), 2B (early-mid pachytene) and 3 (mid pachytene).

in both *c(3)G+* and *c(3)Gmau* ovaries, jointly with immunofluorescence with antibodies against the SC (mau C(3)G) to identify meiotic nuclei and lamin to mark the nuclear boundary. An average of approximately two centromere clusters were observed at all meiotic stages examined for the *c(3)Gmau* nuclei, which was not statistically significantly different from *c(3)G+* ovaries at corresponding stages, indicating the C(3)Gmau can function properly to cluster centromeres (S4 Fig).

As C(3)G is required for the formation of the crossovers that ensure chromosome segregation at the meiosis I division [32], we used a genetic assay to examine the segregation of the *X* and *4th* chromosomes in *c(3)Gmau* females (Table 1). In assays of chromosome segregation, the *c(3)Gmau* females displayed 0.2% *X* and 0.0% *4th* chromosome nondisjunction compared to 0.6% and 0.2% *X* and *4th* chromosome nondisjunction for *c(3)G+* females (Table 1). These levels of non-disjunction are not significantly different and thus, we conclude that sufficient crossovers are being generated to ensure homologous meiosis I chromosome segregation in *c(3)Gmau* females. These results demonstrate the C(3)Gmau protein can substitute for the *D. melanogaster* version of C(3)G in the functions of centromere clustering and meiosis I chromosome segregation.

### DSBs are induced at the correct stage

The SC has been shown to be necessary not only for crossover formation but also for the induction of wild-type levels of DSBs in early pachytene in *D. melanogaster* females [36,53] (Fig 1B). The induction and repair of DSBs were examined using antibodies recognizing γH2Av (which marks DSBs), mau C(3)G (SC), and Orb (a marker of cyst development and oocyte selection). Numerous γH2Av foci could be observed in *c(3)G+* and *c(3)Gmau* ovaries in region 2A in cysts displaying strong Orb expression (*c(3)G+* 12.8 SD+/-2.5 foci compared to *c(3)Gmau* 14.0 SD+/-4.6 foci) (S5 Fig). In both genotypes at region 2B, the average number of γH2Av foci was reduced, indicating DSB repair is properly initiated in *c(3)Gmau* ovaries. The *c(3)Gmau* ovaries did display a greater number of region 2B γH2Av foci (5.1 SD+/- 5.5 foci) compared to the *c(3)G+* control (0.8 SD+/- 2.1 foci) and this was statistically different (p=0.0002). However, this number of region 2B γH2Av foci in the *c(3)G+* ovaries was also less than some previously reported wild-type values for region 2B (3.8 foci for all cysts in region 2B in one study and 8.5 and 3.5 foci for early and late region 2B cysts, respectively, for another) [53,57]. The difference in number of γH2Av foci at region 2B likely does not reflect a defect in DSB repair in the *c(3)Gmau* ovaries since both genotypes displayed less than an average of one remaining focus of γH2Av in region 3 (S5 Fig), and *c(3)Gmau* females displayed wild-type levels of chromosome segregation (Table 1). The small excess in γH2Av foci in region 2A and 2B may reflect a subtle effect on the kinetics of DSB repair due to the C(3)Gmau protein or potentially small differences in the kinetics of cyst development in the newly constructed *c(3)Gmau* stock. The use of the Orb antibody for the DSB assay also allowed for the examination of the degree of activation of the pachytene checkpoint, a meiotic delay that can be induced by defects in the chromosome axis or DSB repair [58,59]. Activation of the pachytene checkpoint leads to a persistence of strong Orb accumulation in the cytoplasm around two nuclei in region 3, the "two-oocyte phenotype" [58,59]. Out of 21 *c(3)Gmau* germaria, 20 displayed a single nucleus with Orb concentrated in the cytoplasm with the remaining germarium

**Table 1. *X* and *4*th chromosome nondisjunction frequency.**

| Genotype[a] | % *X* NDJ[b] | % *4* NDJ | Adj Total[c] |
|---|---|---|---|
| *C(3)G+* | 0.64 | 0.16 | 1878 |
| *C(3)Gmau* | 0.19 | 0.00 | 1043 |

[a] Virgin *y w; c(3)G+ ; svspa-pol (C(3)G+ )* and *y w; c(3)Gmau; svspa-pol (C(3)Gmau)* females were mated to multiple *X^Y, In(1)EN, v f B; C(4)RM, ci eyR* males.

[b] NDJ, nondisjunction.

[c] Adjusted Total is calculated to account for inviable progeny classes (see Materials and methods). Values between the genotypes were not statistically significant as based on [56].

displaying a nucleus with strong Orb accumulation in the cytoplasm and a second nucleus with intermediate Orb concentration. In all 18 of the *c(3)G+* ovaries examined, Orb was concentrated around a single nucleus at region 3. This result indicates the pachytene checkpoint is not induced in *c(3)G^{mau}* ovaries. Overall, these data indicate that *C(3)G^{mau}* can function with the *D. melanogaster* SC components to promote induction of wild-type levels of DSBs at early pachytene and to support DSB repair to the point that γH2AV is absent in the selected region 3 oocyte nucleus.

## Crossover patterning is altered in *c(3)G*mau females and is more similar to *D. mauritiana* than *D. melanogaster*

The chromosome segregation assays described above indicate that a sufficient level of crossovers are formed to ensure proper chromosome segregation. However, we sought to determine if the number and placement of crossovers were affected by the replacement with the C(3)G^{mau} protein. Recombination along the entire *X* chromosome was assayed in both *c(3)G+* and *c(3)G^{mau}* females by scoring visible markers on a multiply marked *X* chromosome. First, we examined the distribution of the types of recovered crossover chromatids. There was a decrease in the portion of non-crossover (NCO) chromatids recovered from *c(3)G^{mau}* compared to *c(3)G+* females (36.5% compared to 47.0%) (Fig 5A and Table 2). This decrease in NCO frequency was accompanied by a proportional increase in the number of double and triple crossover chromatids (DCO and TCO) recovered from *c(3)G^{mau}* females. The fraction of single crossover chromatids (SCO) between the two genotypes was similar (Table 2). The difference between the proportion of NCO/SCO to DCO/TCO classes between *c(3)G^{mau}* and *c(3)G+* was statistically significant (Fisher exact test table p < 0.0001), indicating an excess of chromosomes experiencing more than one crossover.

The rate of crossing over (i.e., the frequency of recombinant offspring divided by the total offspring multiplied by 100) within each interval of the *X* chromosome was compared between *c(3)G+* and *c(3)G^{mau},* revealing increases in crossing over in *c(3)G^{mau}* females for all scored intervals (Fig 5A and Table 2). Fig 5A displays the recombination rate of *c(3)G^{mau}* as a percentage of *c(3)G+* to illustrate the difference in the recombination pattern between the two genotypes. While every interval examined showed a statistically significant increase in crossing over for the *c(3)G^{mau}* females, the most striking increase was in the interval spanning the centromere between *forked (f)* and the wild-type *yellow (y+)* marker translocated to the small right arm of the *X* chromosome (Fig 5A and Table 2), which is the most centromere-proximal interval we examined. Crossing over was increased by 206% in *c(3)G^{mau}* females compared to *c(3)G+* females, indicating that recombination most substantially increased in the centromere-proximal euchromatin of the *X* chromosome. This increase in crossing over in centromere-proximal euchromatin mirrors the strong increase in recombination observed in the centromere-proximal euchromatin of the *D. mauritiana* species [13,27].

To further examine the level of crossing over, Weinstein Exchange Ranks were calculated using Weinstein tetrad analysis for *c(3)G^{mau}* and *c(3)G+* [27,60]. As only one of the four chromatids can be recovered in the progeny during female meiosis, the actual proportion of chromatids with 0,1,2, or more crossovers must be mathematically derived from the recovered progeny and these exchange classes are classified as E0, E1, E2, and E3 respectively. (https://simrcompbio.shinyapps.io/CrossoverMLE/)[27]. For the *X* chromosome the estimated E0 frequency for *c(3)G^{mau}* was 0.0793 compared to 0.0967 for *c(3)G+* (Table 2). The biggest shift was in the calculated E2 class with values of 0.5139 for *c(3)G^{mau}* and 0.297 for *c(3)G+*. The values for *c(3)G^{mau}* females share similarities with Weinstein tetrad frequencies for the *X* chromosome observed in whole-genome sequencing studies in *D. mauritiana* [27]. The estimated *X* chromosome E0, E1, and E2 values for *D. mauritiana* from that study were 0.1048, 0.3238, and 0.5714, respectively [27]. The *X* chromosome Weinstein tetrad frequencies estimated from previous *D. melanogaster* whole-genome sequencing data were 0.1224, 0.6122, and 0.2654 for E0, E1, and E2, respectively [5,27]. The *c(3)G+* control matches well with the whole genome sequencing data of *D. melanogaster* while the *c(3)G^{mau}* results are much more similar to the values observed for *D. mauritiana* [5,27]. The Weinstein tetrad frequencies further illustrate the increase in crossing over in *c(3)G^{mau}* females and that this increase has similarities to the crossover pattern of *D. mauritiana* females for the *X* chromosome.

PLOS Genetics

**A**

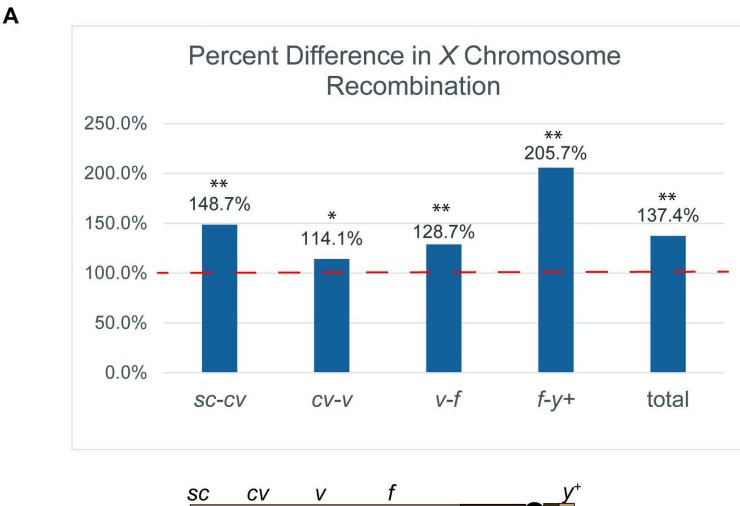

**B**

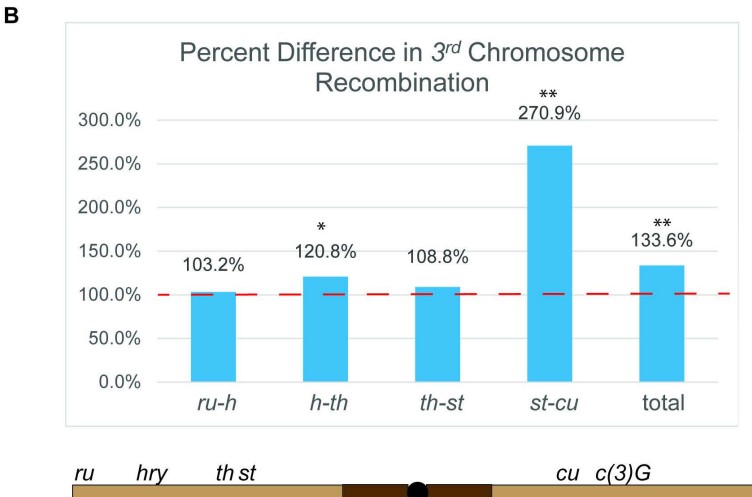

**Fig 5. Recombination in the interval spanning the centromere is increased in *c(3)Gmau* females.** (A) The percent difference in *X* chromosome recombination observed for *c(3)G^mau* compared to the *c(3)G⁺* control (Red dotted line set to 100% of control) for the indicted intervals. Arrangement of the markers scored on the *X* chromosome are diagramed below the graph. (B) The percent difference in *3rd* chromosome recombination observed for *c(3)G^mau* compared to the *c(3)G⁺* control (Red dotted line set to 100% of control) for the indicted intervals. Arrangement of the markers scored on the *3rd* chromosome, as well as the location of *c(3)G*, are diagramed below the graph. Statistically significant from *c(3)G⁺* at * p < 0.05 and **p < 0.01.

Previous studies examining *D. melanogaster* females with internal deletion mutations in the C(3)G protein have demonstrated that the recombination pattern of the *X* chromosome and an autosome can be altered in differing ways [43]. Additionally, while *D. mauritiana* showed greater amounts of crossing over and a decreased centromere effect compared to *D. melanogaster*, the magnitude of those differences were not uniform across all five chromosome arms that experience crossing over [13,27]. We next examined crossing over on the *3rd* chromosome from the marker *roughoid* (*ru*) near the telomere of *3L* to *curled* (*cu*) which is located across the *3rd* chromosome centromere on *3R*. Like the *X* chromosome, there was an increase in DCO chromatids (10.6% for *c(3)G^mau* compared to 5.2% for *c(3)G⁺*) at the expense of NCO chromatids in *c(3)G^mau* (41.7%) compared to *c(3)G⁺* (53.8%) (Table 3). As seen for the *X* chromosome, the difference in the

**Table 2.** *X* Chromosome recombination.

| Maternal Genotype[a] | C(3)G+ | C(3)G^mau |
|---|---|---|
| N[b] | 2626 | 1549 |
| **Map length[c] (% compared to c(3)G+)** | | |
| sc-cv | 13.6 | 20.3 (149%)** [f] |
| cv-v | 20.8 | 23.8 (114%)* |
| v-f | 19.5 | 25.0 (129%)** |
| f-y+ | 7.3 | 15.0 (206%)** |
| Total | 61.2 | 84.1 (137%)** |
| **Class (% of total)[d]** | | |
| NCO | 1233 (47.0%) | 565 (36.5%) |
| SCO | 1182 (45.0%) | 689 (44.5%) |
| DCO | 207 (7.9%) | 271 (17.5%) |
| TCO | 4 (0.2%) | 24 (1.5%) |
| **Exchange rank[e]** | | |
| $E_0$ | 0.0967 | 0.0794 |
| $E_1$ | 0.5941 | 0.2828 |
| $E_2$ | 0.2970 | 0.5139 |
| $E_3$ | 0.0122 | 0.1239 |

[a] Full genotypes are *y¹ sc¹ cv¹ v¹ f¹ y+/y w;; c(3)G+ ; sv^{spa-pol}/+* and *y¹ sc¹ cv¹ v¹ f¹ y+/y w;; c(3)G^{mau}; sv^{spa-pol}/ +*.

[b] Total number of female progeny scored.

[c] The number of recombinants in the interval divided by the total number of progeny scored multiplied by 100.

[d] NCO, chromatids recovered exhibiting no crossovers; SCO, single crossover chromatids; DCO, double crossover chromatids; TCO, triple crossover chromatids.

[e] Calculated using Shiny app https://simrcompbio.shinyapps.io/CrossoverMLE/.

[f] Statistically significant from *c(3)G+* at * $p < 0.05$ and ** $p < 0.01$.

proportion of NCO/SCO chromatids compared to DCO/TCO chromatids was statistically significant for *c(3)G^mau* compared to *c(3)G+* females (Fisher exact test table $p < 0.0001$).

The level of recombination within each of the tested intervals was examined. For the most centromere distal interval, *ru-h,* and the small *Diap1(th)-st* interval recombination levels were similar between *c(3)G^mau* and *c(3)G+* (Fig 5B and, Table 3). A small, but statistically significant, increase in recombination was observed for the *h-Diap1(th)* interval in the medial part of the *3L* chromosome arm for *c(3)G^mau* compared to *c(3)G+* females (Fig 5B and Table 3). What was most noticeable was the 271% increase in crossing over in the *st-cu* interval for *c(3)G^mau* compared to *c(3)G+* (Fig 5B and Table 3). This interval encompasses the centromere and the centromere-proximal euchromatin on both sides of the centromere (Fig 5B). This strong increase in recombination in the centromere-spanning intervals on both the *X* and *3rd* chromosomes suggests a general weakening of the spread of the centromere effect into the centromere-proximal euchromatin in the *c(3)G^mau* females. In our recent recombination study on *D. mauritiana*, two of the regions with the most significant increase in crossing over compared to *D. melanogaster* were the centromere-proximal regions of the *3rd* chromosome [27].

To better evaluate the decrease in NCO chromatids, Exchange Ranks were again calculated using Weinstein tetrad analysis. Similar to the *X* chromosome, *c(3)G^mau* females displayed a decrease in the calculated E0 frequency and an increase in E2 values compared to the control (*c(3)G^mau* E0 = 0.0459 and E2 = 0.3877 compared to *c(3)G+* E0 = 0.1783 and E2 = 0.1676) (Table 3).

**Table 3. 3rd chromosome recombination.**

| Maternal Genotype[a] | C(3)G+ | C(3)Gmau |
|---|---|---|
| N[b] | 1862 | 1589 |
| **Map length (% compared to c(3)Gmel)** | | |
| ru-h | 24.8 | 25.6 (103%) [e] |
| h-th | 19.1 | 23.0 (121%)* |
| th-st | 0.8 | 0.8 (109%) |
| st-cu | 7.4 | 20.1 (271%)** |
| Total | 52.0 | 69.5 (134%)** |
| **Class (% of total)[c]** | | |
| NCO | 1001 (53.8%) | 662 (41.7%) |
| SCO | 759 (40.8%) | 753 (47.4%) |
| DCO | 96 (5.2%) | 169 (10.6%) |
| TCO | 6 (0.3%) | 5 (0.3%) |
| **Exchange rank[d]** | | |
| $E_0$ | 0.178 | 0.046 |
| $E_1$ | 0.628 | 0.541 |
| $E_2$ | 0.168 | 0.388 |
| $E_3$ | 0.026 | 0.025 |

[a] Full genotypes are y w/ +;; ru¹ hry¹ Diap1¹(th) st¹ cu¹ sr¹ eˢ ca¹/ +; svˢᵖᵃ⁻ᵖᵒˡ/+ and y w/ +;; ru¹ hry¹ Diap1¹(th) st¹ cu¹ c(3)Gmau/c(3)Gmau; svˢᵖᵃ⁻ᵖᵒˡ/ +.

[b] Total number of female progeny scored.

[c] NCO, chromatids recovered exhibiting no crossovers; SCO, single crossover chromatids; DCO, double crossover chromatids; TCO, triple crossover chromatids.

[d] Calculated using the Shinyapp at https://simrcompbio.shinyapps.io/CrossoverMLE/.

[e] Statistically significant from c(3)G+ at * $p < 0.05$ and ** $p < 0.01$.

Decreased recombination, especially when severe, is associated with chromosome missegregation, which ultimately can affect fertility due to the production of aneuploid offspring. Whether the increased total recombination or increased centromere-proximal crossovers adversely affected fertility was examined. Prior analyses of the less common meiosis II chromosome missegregation events found correlations with the increased incidence of centromere proximal crossovers (flies and humans) and increased total recombination (humans) [61,62]. Additionally, Arabidopsis lines with increased levels of crossing over display mild decreases in fertility (7–12%) and a low level of aneuploid offspring [48]. In the *X* recombination assay c(3)G+ control females produced an average of 104.5 progeny per scored female while the c(3)Gmau females produced an average of 139.8 progeny per scored female. This result, along with the *X* chromosome missegregation data, suggests that the increased level of crossing over in the centromere proximal euchromatin in c(3)Gmau females does not impact fertility or chromosome segregation. Altogether, our results indicate that the presence of C(3)Gmau protein in *D. melanogaster* females can support normal SC functions and promotes increased crossing over on multiple chromosome arms and generates a crossover pattern more similar to the *D. mauritiana* species than wild-type *D. melanogaster* females.

## Discussion

### *Cis* versus *trans*-acting factors regulating changes in crossover patterning

The number of crossovers formed per bivalent and their location on the chromosomes can vary between species, leading to species-specific crossover patterns that can differ even between closely related species. It is not fully understood

how crossover patterns can evolve. Are new crossover patterns driven by changes in just a few genes or are crossover patterns derived from the small effects of many genes and cis-acting genetic elements?

Here, we show that complete replacement of the *D. melanogaster c(3)G* gene with *D. mauritiana c(3)G* led to the alteration of the recombination pattern of *D. melanogaster* to look more like the *D. mauritiana* crossover pattern, with increased crossing over in the centromere-proximal euchromatin and an increase in total crossing over for the *X* and *3rd* chromosomes. Examining the calculated E0 frequencies from these experiments revealed a decrease in NCO chromatids in *c(3)Gmau* females. The *c(3)Gmau* females displayed wild-type levels of chromosome segregation and fertility, supporting that the changes observed in crossover placement and number were not detrimental to meiosis. Given that this substitution represents a single gene, the results suggest that global crossover patterning can be strongly influenced by a relatively small number of genetic loci and supports the inference that the C(3)G protein and/or the SC is a mechanism to regulate crossover patterning in *Drosophila*.

*c(3)G* is not the first example of a single gene noticeably altering crossover patterning in *D. melanogaster*. Brand *et al.* [28] expressed the *D. mauritiana* version of the pro-crossover protein Mei-218 as a transgene in *D. melanogaster* that lacked a functional endogenous *mei-218* gene. Expression of *D. mauritiana mei-218* increased total crossing over, particularly for genetic intervals near the telomere and centromeres. However, in that study, it was not apparent that a wild-type genetic map could be fully restored, with either the *D. mauritiana* or *D. melanogaster* transgenes. In comparison to the *mei-218* transgene results, the *c(3)Gmau* replacement had a lesser impact on regions near the telomere with the increases in recombination most focused on intervals spanning the centromeres.

Unlike studies directly comparing recombination between the *D. melanogaster* and *D. mauritiana* species, where the centromere and centromere-proximal heterochromatin are different [13,27], in the *mei-218* and *c(3)G* studies the *cis*-acting elements at or near the centromere should be essentially identical between the compared flies. Therefore, the changes observed in crossover patterning within the *D. melanogaster* females could only be driven by the *trans*-acting factors that were manipulated in the studies. This strongly suggests that the difference in the strength of the centromere effect between *D. melanogaster* and *D. mauritiana* is primarily mediated by *trans*-acting factors, rather than specific *cis*-acting elements within the centromeres and surrounding heterochromatin of those species. *Cis* acting elements, such as heterochromatin and hotspots, would still influence local placement of crossovers but these *trans*-acting factors apparently enable global changes in patterning that are observed across species.

Why were Mei-218 and C(3)G good candidates for regulating aspects of global crossover patterning? Both are rapidly evolving proteins [28,38,39] and are involved in the formation of crossovers in *D. melanogaster* [32,63]. While genes involved in meiosis tend to show higher rates of evolution than genes that function in the soma, Mei-218 was chosen to test because it was found to be particularly fast evolving [28]. C(3)G and other components of the SC are evolving so rapidly that they cannot readily be identified outside the genus based on amino acid sequence alone [38,39]. The rapid divergence of these proteins makes both C(3)G and Mei-218 good candidates for driving rapid evolution of the recombination landscape. If evolution of the recombination landscape occurred only by *cis*-acting factors, rapid evolution of the global landscape would require many changes occurring in a short span of time, which seems less plausible.

One thing to consider, however, is whether the mechanism, by which replacement of the native *D. melanogaster* C(3)G with that of *D. mauritiana* leads to a change in the recombination landscape (to one more similar to *D. mauritiana*), is identical to the cause of divergence in the recombination landscape between these two species. In fact, the observed transformation to a *D. mauritiana*-like recombination landscape may instead occur through an alternative mechanism, and the similarity may be a case of mere coincidental similarity. For example, if there is some cryptic disruption in SC function that occurs in the *D. melanogaster* replacement line, but in neither of the two species, this could trigger some event that leads to a change in recombination through an alternative mechanism. We do not believe this is likely to be the case, because the SC phenotype was exhaustively investigated and only marginal differences were found. Still, we cannot completely rule out the possibility that the similarity of the recombination landscape to that of wild-type *D. mauritiana* in the replacement line is through a different mechanism than the one that causes the differences between the two species.

## The role of the SC in crossover patterning

How might the SC alter recombination patterns? Current models predict that pro-crossover factors can diffuse within the SC and then, through a coarsening process, accumulate at the subset of DSBs that become designated to become crossovers [46–48]. It was recently shown that a mutation in the *C. elegans* SC component *syp-4*, promoted SC assembly (based on standard immunofluorescence), but it was also associated with an increased number of crossovers compared to wild-type worms [44]. Examination of the SC by three-dimensional stochastic optical reconstruction microscopy (3D-STORM) revealed changes in the organization of the components of the SC in the *syp-4* mutant [44]. Internal deletion mutations in c(3)G cause defects in SC assembly and/or disassembly, as well as changes in crossover levels and placement [43]. More recently, it was shown that there is an increase in crossovers in the β-heterochromatin associated with the $2^{nd}$ chromosome centromere in one of these deletion mutants, indicating C(3)G and/or the SC plays a role in suppressing crossovers within parts of the centromeric heterochromatin [12].

Our work here demonstrated that the *C(3)G^mau* replacement can promote full-length tripartite SC assembly, chromosome segregation, fertility, and centromere clustering, indicating this version of C(3)G can interact well enough with *D. melanogaster* meiotic components to promote these important functions. The initiation of premature SC disassembly was observed in about a quarter of germaria at region 3 (mid pachytene), but this phenotype is mild compared to the phenotype of *c(3)G* internal deletion mutants [43] and occurs after crossovers are designated based on the localization of the RING domain protein Vilya [52]. While tripartite SC structure was observed by super-resolution microscopy in *c(3)G^mau*, a statistically significant decrease in width between the tracks of mau C(3)G antibody fluorescence was also observed. The difference in width may contribute to differences in SC function. Along with the mild disassembly phenotype, the difference in SC width opens the possibility there may be some difference in the organization of the SC components in *c(3)G^mau* ovaries which could lead to an SC that is partially dysfunctional. If use of the C(3)G^mau protein caused changes in the interaction of SC proteins similar to the recently described *syp-4* mutant in *C. elegans* [44], the altered, and potentially dysfunctional, SC structure could change how pro-crossover factors diffuse within the SC to ultimately affect the number and location of DSBs designated to become crossovers. However, as previously described, most *D. melanogaster* SC mutations lead to decreases in crossing over in at least some genetic intervals suggesting that any SC dysfunction caused by use of C(3)G^mau is likely different from the C(3)G deletion mutants [43]. In studies of complete reconstructions of electron microscopy data of nuclei from fly germaria, Carpenter [64] described the SC of the heterochromatin approaching the chromocenter as having a different morphology than the SC of euchromatin, with less distinct LEs and a relatively amorphous central element. The recent identification of crossovers in the β -heterochromatin in a *c(3)G* deletion mutant suggests that the morphological difference in SC structure could potentially be part of the mechanism of the centromere effect in *D. melanogaster,* perhaps by varying the manner in which centromeric suppression is transmitted along the chromosome arm [12]. It would be interesting to use EM in the future to explore whether the C(3)G^mau protein could potentially be altering the organization of the SC in and near the centromere-proximal heterochromatin. One can imagine a change in SC structure in these regions could make recruitment of pro-crossover factors to the euchromatic regions closer to the centromeres more likely and increase crossing over in the centromere-proximal euchromatin.

The SC was observed in EM images decades earlier (Moses [65]), and its role in promoting crossover formation in many species has been of intense interest. Only more recently have there been studies on mutations or conditions that effect the SC without disrupting the structure entirely. These studies are beginning to reveal additional, influential roles of this conserved structure and are providing information on how crossover patterns may be regulated and altered by SC evolution.

## Materials and methods

### Drosophila husbandry

Flies were maintained at 25°C on standard food. *c(3)G^mau* genotype is *y w/ y + Y; (3)G^mau; sv^spa_pol* except for recombination experiments (see Recombination methods). Control (*c(3)G^+*) stock was *y w/ y + Y; sv^spa_pol* to be genetically similar to the *c(3)G^mau* stock, with the exception of recombination experiments (see Recombination methods).

The replacement of the *c(3)G* gene in *D. melanogaster* was performed by WellGenetics using CRISPR. A donor template containing a *D. melanogaster* codon optimized coding region of the *D. mauritiana c(3)G* was constructed. Codon optimization was performed using the IDT DNA codon optimization tool (https://www.idtdna.com/CodonOpt) with the gBlocks option (S1 Fig). Codon optimization was used to ensure robust expression of the *D. mauritiana c(3)G* gene, and by varying many of the synonymous positions, to limit difficulties of homologous replacement using CRISPR. CRISPR-mediated mutagenesis for the codon-optimized *D. mauritiana c(3)G* replacement was performed by WellGenetics Inc. using modified methods described in [66]. Briefly, the upstream gRNA sequence ACGGGCGC-CAAGAAAAGATC[GGG] and the downstream gRNA sequence CTTGATGGGCGCACATAATT[TGG] were cloned into U6 promoter plasmids. The cassette Dmau c(3)G cDNA-PBacDsRed, containing 3xP3-DsRed flanked by PiggyBac terminal repeats, and two homology arms were cloned into pUC57-Kan as donor template for repair. *c(3)G/CG17604*-targeting gRNAs and hs-Cas9 were supplied in DNA plasmids, together with donor plasmid, for microinjection into embryos of control strain *[SWGa3411] w^{1118} hyb5b-8*. F1 flies carrying the 3xP3-DsRed selection marker were further validated by genomic PCR and sequencing. The CRISPR generated a 2,655-bp deletion of *c(3)G/CG17604,* which deleted the entire CDS of *c(3)G/CG17604* that was replaced by cassette Dmau c(3)G cDNA-PBacDsRed. The 3xP3-DsRed cassette was then excised through PiggyBac transposition, and the excision was validated by PCR and sequencing.

The *c(3)G^{mau}* gene was integrated into the *y w/ y + Y; sv^{spa_pol}* genetic background using standard genetic crosses and the replacement PCR verified.

## Immunofluorescence

For immunofluorescence of whole-mount ovaries, females 1–3 days post-eclosion were provided with wet yeast paste and males overnight at 25°C. Ovaries were dissected in PBS plus 0.1% Tween-20 (PBST). Ovaries were fixed for 20 min in 200 μL of PBS containing 2% formaldehyde (Ted Pella) and 0.5% Nonidet P-40 plus 600 μL heptane. After three washes in PBST, ovaries were blocked for at least one hour in PBST plus 1% bovine serum albumin (BSA) (EMD Chemicals, San Diego, CA). For experiments using the γH2Av antibody, the primary antibodies were applied in PBST plus 0.5% BSA over-night at 4°C, while for all other experiments primary antibodies were applied in PBST overnight. For experiments using the γH2Av antibody, ovaries were washed three times in PBST before blocking again for at least one hour in PBST plus 1% BSA. Secondary antibodies were applied overnight at 4°C in PBST plus 0.5% BSA. For all other experiments, after the three washes in PBST, secondary antibodies were applied for 3–5 hours at room temperature. For all experiments, 1.0 μg/mL DAPI was applied during the last 15 minutes of secondary antibody incubation. Samples were washed three times in PBST. Ovaries were mounted in Prolong Gold or Prolong Glass (Thermofisher).

Primary antibodies rat anti- *D. mauritiana* C(3)G (see Antibody Production methods), rabbit anti-histone γH2AVD pS137 [1:5000] (Rockland Inc., Lot 46042), mouse anti-Orb 6HA [1:40] (Developmental Studies Hybridoma Bank, Iowa) [67], rabbit anti-Corolla [1:2000] [36], rabbit anti-Centromere Identifier (CID) [1:2000] (gift of Dr. Gregory Rogers), mouse anti-C(3)G *D. melanogaster* IA8-IG2 [1:500] [32], and mouse anti-lamin Dm0 ADL84.12 [1:100] (Developmental Studies Hybridoma Bank, Iowa) were used. The following goat Alexa Fluor secondary antibodies at [1:500] were used: anti-mouse 488 or 647, anti-rat 488, 555, and 594, and anti-rabbit 488 or 555 (ThermoFisher).

## Microscopes and imaging conditions

For most images, a Deltavision Elite system from GE Healthcare equipped with an Olympus IX70 inverted microscope and a high-resolution CCD camera was used. SoftWoRx v. 7.2.1 (Applied Precision/Leica Microsystems, Inc,) was used to deconvolve images. SoftWoRx v. 7.2.1 or Fiji were used for image analysis. Brightness and contrast were adjusted minimally for improved publication quality.

The STED imaging in Fig 2 was performed using a Leica SP8 Gated STED Microscope equipped with a 100×, 1.4 NA oil immersion objective. The C3G channel (labeled with Alexa 594) was excited using a pulsed white light source (80 MHz) set to 594 nm, in combination with a pulsed 775 nm laser operating at 80–90% of its maximum output. All STED images

were captured in 2D mode to optimize lateral resolution, with each image averaged eight times in line average mode. Emission photons were detected using internal Leica HyD hybrid detectors, with a time gate setting of 1–6 ns. Raw STED images were subsequently deconvolved using Huygens Professional software (version 14.10; Scientific Volume Imaging), employing a theoretically estimated point spread function for the deconvolution process. Default system parameters were used, except for the background, which was measured directly from the raw images, and the signal-to-noise ratio, which was adjusted to a range of 15–20.

To measure the distance between tracks of the C-terminal mau C(3)G antibody in STED images, line profile measurements were acquired from line ROIs drawn manually perpendicularly across regions of the SC that appeared to be flat in the z dimension. The average and standard deviation for the measurements were calculated for the measurements in Excel and a Mann-Whitney test was used for statistical comparisons between the distances for $c(3)G^+$ and $c(3)G^{mau}$.

## Antibody production

For the production of the *D. mauritiana* C(3)G antibody a construct encoding 324 amino acids in the C-terminal part of the protein (NESYLNELTETKLKHTQEIKEQADAYEIVVQELKESLNKASVDFTQLKSNSEKLHKETLLQVSQLQEKLTE MVSHRSNQEELVKRLGDELQEKTHSFEEELKRQQVQLANQMQMKTTEVASENKRNAVQIQTLKSELEDRNKAF-KAQQDKLEFLISDLDKLKNAIINLQAEKMEIESELSTAKVKFSEELQFQKDSLMKKVSELELEIKRKENELIELERE-KNNEMAVLQFKMNRINCVIDQPVTTIKQLKDSKGPTKTESIPTKVQPENTDGFSGTAKKRNARRQKITTYSSDFD SGDDMPILGNSLNGKRVKLCTPIKPQNN) (See S2 Fig) with a start codon and N-terminal His tag was made in a pET30a by GenScript USA, Inc. The vector was expressed in BL21 (DE3) bacterial cells, and the protein was purified using Ni-NTA His-Bind Resin (EMD Millipore Corp) under denaturing conditions. The purified protein was used to generate an antibody in rats by Cocalico Biologicals, Inc.

## Recombination and fertility assays

For recombination studies of the *X* chromosomes virgin $y^1 sc^1 cv^1 v^1 f^1 y +/y$ *w;; sv*$^{spa\_pol}$/+ $(c(3)G^+)$ or $y^1 sc^1 cv^1 v^1 f^1 y +/y$ *w;; c(3)G*$^{mau}$; *sv*$^{spa-pol}$/+ $(c(3)G^{mau})$ females were crossed individually to *y sc cv v f car*$^1$/ $B^SY$ males. Only female progeny were scored for the genetic markers. Females were brooded once and progeny from the vials were scored for up to 18 days. For fertility analysis, the total number of female and male progeny were divided by the number of females producing progeny.

For *3*$^{rd}$ chromosome recombination scoring a recombinant *ru hry Diap1*$^1$ *(th) st cu c(3)G*$^{mau}$ stock was constructed using standard genetic methods. Virgin *y w/ +;; ru*$^1$ *hry*$^1$ *Diap1*$^1$*(th) st*$^1$ *cu*$^1$ *sr*$^1$ *e*$^s$ *ca*$^1$/ +; *sv*$^{spa-pol}$/+ $(c(3)G^+)$ or *y w/ +;; ru*$^1$ *hry*$^1$ *Diap1*$^1$*(th) st*$^1$ *cu*$^1$ *c(3)G*$^{mau}$/*c(3)G*$^{mau}$; *sv*$^{spa-pol}$/+ $(c(3)G^{mau})$ females were mated individually to *ru*$^1$ *hry*$^1$ *Diap1*$^1$*(th) st*$^1$ *cu*$^1$ *sr*$^1$ *e*$^s$ *ca*$^1$ males. Only female progeny were scored for the markers *ru hry Diap1*$^1$ *(th) st cu*. Females were brooded once and progeny from the vials were scored for up to 18 days.

## Meiotic nondisjunction

To measure the rate of chromosome nondisjunction for both the *X* and *4th* chromosomes, virgin females of the indicated genotypes were mated to multiple *X^Y, In(1)EN, v f B; C(4)RM, ci ey*$^R$ males. The calculations to determine the percentage of *X* and *4th* chromosome nondisjunction, as well as adjusted progeny total, are described in [56,68,69]. Statistics test is described in [56].

## Statistics and tetrad calculations

Fisher's exact test was used to compare the number of NCO/SCO to DCO/TCO numbers between genotypes and the difference for each interval in recombination experiments (https://www.graphpad.com/quickcalcs/contingency1/).

Mann-Whitney U test was used to compare genotypes for centromere clustering, γH2Av foci number, and the distance between the mau C(3)G fluorescence (https://miniwebtool.com/mann-whitney-u-test-calculator/). The maximum likelihood estimates (MLE) of tetrad classes were calculated using the shiny app https://simrcompbio.shinyapps.io/CrossoverMLE/ [27].

### Analysis of SC assembly and disassembly and γH2Av dynamics

Germaria stained with antibodies for mau C(3)G (SC), γH2Av (DSBs), and Orb (oocyte development) were analyzed for DSB and SC dynamics. For region 2A, only SC positive nuclei from cysts with strong Orb expression throughout the cyst were scored. Nuclei from different parts of 2A were scored, and typically more than one region 2A nucleus was analyzed for each germarium. For region 2B, if Orb had accumulated around a single nucleus in a cyst, that was the nucleus scored. If more than one cyst was present in region 2B, a nucleus was scored for DSB number from both cysts if the nuclei were unobstructed. For region 3, the Orb selected nucleus was scored for DSBs. To score DSB number, the nuclei of interest was duplicated in X, Y, and Z in Fiji and the individual Z slices made into a montage using a Make Montage Plugin. The γH2Av foci were then counted as they appeared through the Z slices.

For SC assembly state, the scored nuclei were described as one of four categories: 1) full-length with long, continuous tracks of SC; 2) mild fragmentation with some long tracks of SC and some partial tracks of SC; 3) highly fragmented where C(3)G is present but no long tracks of SC were observed; and 4) absent, in that there is a complete lack of SC structures in Orb selected nuclei (no nuclei fell into this category). Only one value was assigned for each region of each germarium for SC assembly/disassembly. In region 2B, if the two cysts had differing SC classifications, the classification assigned was the more mature region 2B cyst.

## Supporting information

**S1 Fig. Codon optimized *D. mauritiana c(3)G* coding sequence.** The codon optimized nucleotide *D. mauritiana c(3)G* coding sequence (line 1) is shown aligned with publicly available nucleotide sequences for *D. mauritiana* (line 2) and *D. melanogaster c(3)G* (line 3). Amino acid sequence coded by each codon is also displayed illustrating that the codon optimized nucleotide sequence retained the amino acid sequence of the publicly available *D. mauritiana* sequence.
(PDF)

**S2 Fig. Alignment of D. *melanogaster* and *D. mauritiana* C(3)G proteins.** Amino acid differences are displayed in red and gaps in white. The proteins show 89.1% identity with differences dispersed throughout the protein. The C(3)G protein is predicted to have a large coiled-coil domain in the center of the protein. The predicted start and end of the coiled-coil domain is indicated on the *D. melanogaster* sequence by pink triangles. Green and purple lines just above the *D. melanogaster* sequence indicate the locations of the three deletion mutations described in Billmyre et al. [43]. Asterisks below the *D. mauritiana* sequence indicate the start and end of the region expressed to generate the *D. mauritiana* C(3)G antibody.
(PDF)

**S3 Fig. The antibody recognizing the C-terminus of *D. melanogaster* C(3)G fails to recognize *D. mauritiana* C(3)G.** Germaria from *c(3)G⁺* (top) and *c(3)G^{mau}* (bottom) stained with antibodies recognizing the C-terminus of the *D. melanogaster* C(3)G protein (green) and Corolla (magenta).While the antibodies show overlapping localization in *c(3)G⁺*, the *D. melanogaster* C(3)G antibody does not show colocalization with Corolla in *c(3)G^{mau}*. Images are projections from z-stacks and scale bar = 10 μm.
(TIF)

**S4 Fig. Centromeres cluster properly in *c(3)G*mau ovaries.** (A) Examples of nuclei from *c(3)G^{mau}* ovaries with 1 (top row), 2, or 3 (bottom row) centromere clusters. Centromeres are labeled with an antibody recognizing the centromeric

histone CID (magenta), mau C(3)G antibody labels the SC (yellow), and lamin antibody outlines the nuclear envelope (cyan). Asterisks indicate the location of centromere clusters. Scale bar = 1 µm and images are projections from partial z-stacks. (B) The average number of centromere clusters with standard deviation at the indicated developmental stages in *c(3)G⁺* and *c(3)Gᵐᵃᵘ* ovaries. Number of nuclei scored in parentheses. Mann-Whitney U test found no statistically significant difference between the genotypes for any of the stages examined (p > 0.05).
(TIF)

**S5 Fig. Double strand breaks are induced at early pachytene and undergo repair in *c(3)G*mau ovaries.** (A) *c(3)G⁺* (top) and *c(3)Gᵐᵃᵘ* (bottom) germaria were stained with antibodies to mau C(3)G to identify the SC (green), γH2Av to recognize DSBs (magenta), Orb to stage oocyte development (white), and DAPI (blue). In both genotypes DSBs are induced in region 2A (early pachytene) and γH2Av staining is absent from the Orb selected nucleus in region 3 (mid pachytene). Note in the *c(3)Gᵐᵃᵘ* image γH2Av foci are present in the nurse cells that have started endoreduplication in region 3. Scale bar = 10 µm and images are projections from z stacks. (B) Table provides the average number with standard deviation of γH2Av foci for each region of the germarium. The total number of nuclei scored is in parentheses. By Mann-Whitney U test the statistically significant differences from *c(3)G⁺* are indicated as * p < 0.05 and **p < 0.01. Full genotypes are *y w;; c(3)G⁺; spaᵖᵒˡ* and *y w;; c(3)Gᵐᵃᵘ; spaᵖᵒˡ*.
(TIF)

## Acknowledgments

Thank you to Cathy Lake, Jay Unruh, Leah Rosin, and Kimla Virden for valuable comments on the manuscript and Stefanie Williams for her help with figure preparation and suggestions. R.S.H. was an American Cancer Society Research Professor.

## Author contributions

**Conceptualization:** R. Scott Hawley.

**Data curation:** Cynthia Staber, Grace McKown.

**Formal analysis:** Stacie E. Hughes, Cynthia Staber, Justin P Blumenstiel, R. Scott Hawley.

**Funding acquisition:** R. Scott Hawley.

**Investigation:** Stacie E. Hughes, Grace McKown, Zulin Yu.

**Methodology:** Cynthia Staber.

**Project administration:** Stacie E. Hughes.

**Resources:** R. Scott Hawley.

**Supervision:** Stacie E. Hughes.

**Writing – original draft:** Stacie E. Hughes, Justin P Blumenstiel.

**Writing – review & editing:** Stacie E. Hughes, Justin P Blumenstiel.

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
