## [Decision Letter · Decision Letter 0]

16 Jul 2025

PGENETICS-D-25-00642

The recombination landscape of Drosophila melanogaster can be repatterned by a single gene

PLOS Genetics

Dear Dr. Hughes,

Thank you for submitting your manuscript to PLOS Genetics. After careful consideration, we feel that it has merit but does not fully meet PLOS Genetics's publication criteria as it currently stands. Therefore, we invite you to submit a revised version of the manuscript that addresses the points raised during the review process. Additional experiments may not be necessary, but make sure that all reviewers comments are addressed. In the absence of additional experiments, please note the specific concerns and suggestions raised by Reviewer 2.

Please submit your revised manuscript within 60 days Sep 14 2025 11:59PM. If you will need more time than this to complete your revisions, please reply to this message or contact the journal office at plosgenetics@plos.org. Please include the following items when submitting your revised manuscript:

We look forward to receiving your revised manuscript.

Kind regards,

Aimee Jaramillo-Lambert, Ph.D.

Guest Editor

PLOS Genetics

Paula Cohen

Section Editor

PLOS Genetics

Aimée Dudley

Editor-in-Chief

PLOS Genetics

Anne Goriely

Editor-in-Chief

PLOS Genetics

**Journal Requirements:**

Please ensure that the funders and grant numbers match between the Financial Disclosure field and the Funding Information tab in your submission form. Note that the funders must be provided in the same order in both places as well.

2) State what role the funders took in the study. If the funders had no role in your study, please state: "The funders had no role in study design, data collection and analysis, decision to publish, or preparation of the manuscript.".

**Reviewers' comments:**

Reviewer's Responses to Questions

**Comments to the Authors:**

Reviewer #1: The manuscript by Hugues et al. explores the consequences of replacing the central element protein C(3G) from D. melanogaster with its ortholog from D. mauritana. The data demonstrate that D. mauritana C(3G) is functional in D. melanogaster and supports all meiotic functions. Interestingly, this results in a significant increase in crossover frequency in pericentromeric intervals, suggesting that C(3)G supports the centromere effect (suppression of CO) and that variations in this protein can generate interspecific differences in crossover distribution.

The manuscript is well organized, and I only have some suggestions to improve clarity.

On line 95, define "centromere" and describe its structure.

Line 103: Clarify/correct "increased level of suppression."

Lines 123–135: Improve the readability of this sentence.

Results: The first two to three paragraphs should be included in the introduction.

Line 218: Clarify/specify "similar genetic background"

L233-234 is unclear. Do you mean that no signal was detected in C(3)Gmau fly oocytes?

L251. Define "n."

L324: "DSB foci" should be "gH2Av foci."

L380 and Tables. What are the "rate of crossing over" and "map length"? Is it simply a fraction of recombined chromatids or another calculation?

L532-533. "A mutation promotes SC assembly." This should be rephrased.

Reviewer #2: This paper, by Stacie Hughes, the late Scott Hawley and colleagues addresses the potential role of C(3)G - the fly transverse filament protein - in regulating the crossover landscape. Specifically, they test the hypothesis that divergence in the amino acid sequence of C(3)G between D. melanogaster and D. mauritania is responsible for differences in the crossover landscape between these two closely related species. They do that by precise gene replacement of C(3)G and analysis of the recombination landscape. They indeed identified a relatively dramatic change in the crossover landscape that is consistent with the differences observed between the the two Drosophila species. Given our still-limited understanding of crossover regulation by the synaptonemal complex, linking a single protein to a specific effect on crossover distribution would constitute an important advance in the field.

The major concern with the manuscript is the conclusion that swapping C(3)G caused the recombination landscape in melanogaster to become more similar to that landscape of Mauritania. While the effect of the swap is consistent with this possibility, there are other potential possibilities. The one most likely in my opinion is that the observed effects are the result of a somewhat malfunctioning SC. This idea is consistent with some similarity with the effect the Hawley lab observed in strains with small deletions in C(3)G (the effect of these small deletions were distinct from the ones observed here, but are not unrelated). It is also consistent with the prolonged duration of H2Av staining. In other words, the observed phenotypes could be the result of some inter-chromosomal effect (analogous to the effect of pairing & synapsis mutants on the CO landscape in worms).

There are two possible ways to resolve this issue. One will be to perform significant amount of additional work to rule out dysfunction of the SC in the swapped strain. That would not be trivial to do, but as a start, more comprehensive analysis to show that the SC is functioning normally (or as normally as could be assessed) would be needed. It will also involve some chasing down of the extended H2Av signal. In addition, the degree of similarity to the D. mauritania CO landscape should be quantified (and, if the claim in relation to the mei-218 swap is maintained, that difference should also be quantified.) The second possibility is to dramatically tone down the claims of this manuscript to better reflect the findings of this paper.

The other concerns are more minor. The title of the manuscript, which does not reflect the novelty of this work, should be edited. As mentioned above, multiple single-gene alterations were shown to affect the recombination landscape. A more descriptive title would be more appropriate. In addition, the text and figures would benefit from significant streamlining. It is more appropriate as a report- or letter-style format. (I realize PLoS Genetics doesn’t formally have such a format, but aiming for the equivalent of that would be ideal.)

Reviewer #3: Over the past ~10 years, the role of the synaptonemal complex in crossover regulation has been an emergent theme among crossover regulation papers across organisms. This manuscript by Hughes et al. takes the approach of replacing the transverse filament SC component C(3)G in Drosophila melanogaster with that of Drosophila mauritiana, in which it is known to have decreased crossover number and patterning. Prior work has shown that replacement of D. melanogaster mei-218 with that of D. mauritiana found that crossovers specifically in the telomeres and centromeres were increasing. By performing this elegant approach with C(3)G, Hughes et al. were able to recapitulate the crossover landscape in D. mauritiana for the intervals they tested across several chromosomes. This work not only emphasizes the importance of the SC and the transverse filament in crossover regulation, but also starts to identify the molecular mechanisms behind this regulation of the SC in crossover regulation (e.g. how the width and specific residues may affect the propagation of an interference ‘signal’ within the SC) as well as solidify the fact crossover regulation (and perhaps interference) is controlled by multiple factors. I only have minor comments:

1. The paper both strongly builds and supports the findings and suggestions of the Billmyre and Cahoon et al., PNAS 2019 paper where they made internal in frame deletions of the coiled-coil domain of D. melanogaster C(3)G and found specific effects on recombination for each deletion. The conclusions and interpretations of the present study would be strengthened by indicating where the amino acid differences in C(3)G between D. melanogaster and D. mauritiana map to the internal deletions made in the Billmyre 2019 paper. Figure S2 or an additional figure could be generated to indicate where the structure domains of C(3)G are in the two a.a. sequences and where the Billmyre deletions map at least in the melanogaster sequence.

2. Whether decreased (or a loss) of crossover interference is required for fertility is an ongoing debate (currently thought to not necessarily be required for fertility), therefore reporting the overall effect on fertility (e.g. brood size and dead egg/embryo) of C(3)G of D. mauritiana in the D. melanogaster where crossover interference is now decreased may increase in the ability of the paper to be cited in the context of future studies that resolve this question of whether the highly conserved process of crossover interference is required for fertility.

3. Figure S3 - The study finds that the SC is destabilized in specific stages of the meiosis (e.g. regions of the germaria). Specifically, they find that region 3 (mid-pachytene) may be affected. I think this is an important finding about the regulation/stability of the SC and how specific residues of the transverse filament (or the width of the SC) may impact this. I think Figure S3 should be added to the main figures of the paper so that this finding isn’t missed by readers. For Figure S3, the addition of D. melanogaster example images of the different stages/regions as well as state to indicate the significances of the difference would strength the figure for addition to the main figures.

4. The study finds that the SC width with the C(3)G of D. mauritiana is thinner than that of D. melanogaster, which is commiserate with a decrease in crossover interference especially in centromere proximal regions. This is a very minor point (especially given differences in genome size, centromere-type, heterochromatin), but how does this specific SC width and interference strength compare with other organisms that have a similar SC width?

**Have all data underlying the figures and results presented in the manuscript been provided?**

Reviewer #1: None

Reviewer #2: Yes

Reviewer #3: Yes

PLOS authors have the option to publish the peer review history of their article (what does this mean? ). If published, this will include your full peer review and any attached files.

**Do you want your identity to be public for this peer review?** For information about this choice, including consent withdrawal, please see our Privacy Policy .

Reviewer #1: No

Reviewer #2: No

Reviewer #3: No

**Figure resubmission:**
---

## [Decision Letter · Decision Letter 1]

13 Sep 2025

Dear Stacie,

We are pleased to inform you that your manuscript entitled "The Drosophila mauritiana synaptonemal complex protein C(3)G repatterns the recombination landscape of Drosophila melanogaster" has been editorially accepted for publication in PLOS Genetics. Congratulations!

Yours sincerely,

Aimee Jaramillo-Lambert, Ph.D.

Guest Editor

PLOS Genetics

Paula Cohen

Section Editor

PLOS Genetics

Aimée Dudley

Editor-in-Chief

PLOS Genetics

Anne Goriely

Editor-in-Chief

PLOS Genetics

Comments from the reviewers (if applicable):

Reviewer's Responses to Questions

**Comments to the Authors:**

Reviewer #1: The issues raised have been properly addressed by the authors. I have no further reservations.

Reviewer #2: I am content with the changes to the two main issues I raised in the first round of review. I find the chance that an “alternative mechanism Q” is affected more likely than the authors. Specifically, I think it is likely that some incompatibility between C(3)g and one of its interacting partners might be at play. That said, the changes the authors introduced cover that possibility, leaving both options open and to-be-addressed in future studies.

Reviewer #3: The authors have done a beautiful job at addressing all of the reviews. It is acceptable for publication.

**Have all data underlying the figures and results presented in the manuscript been provided?**

Reviewer #1: Yes

Reviewer #2: Yes

Reviewer #3: Yes

PLOS authors have the option to publish the peer review history of their article (what does this mean? ). If published, this will include your full peer review and any attached files.

**Do you want your identity to be public for this peer review?** For information about this choice, including consent withdrawal, please see our Privacy Policy .

Reviewer #1: No

Reviewer #2: No

Reviewer #3: No

**Data Deposition**

http://datadryad.org/submit?journalID=pgenetics&manu=PGENETICS-D-25-00642R1

**Press Queries**

---

## [Editor Report · Acceptance letter]

PGENETICS-D-25-00642R1

The Drosophila mauritiana synaptonemal complex protein C(3)G repatterns the recombination landscape of Drosophila melanogaster

Dear Dr Hughes,

We are pleased to inform you that your manuscript entitled "The Drosophila mauritiana synaptonemal complex protein C(3)G repatterns the recombination landscape of Drosophila melanogaster" has been formally accepted for publication in PLOS Genetics! Your manuscript is now with our production department and you will be notified of the publication date in due course.

With kind regards,

Olena Szabo

PLOS Genetics

On behalf of:
